

# Characterization of a Mn-SOD from the desert beetle *Microdera punctipennis* and its increased resistance to cold stress in *E. coli* cells

Zilajiguli Xikeranmu, Ji Ma and Xiaoning Liu

Xinjiang Key Laboratory of Biological Resources and Genetic Engineering, College of Life Science and Technology, Xinjiang University, Urumqi, China

## ABSTRACT

Insects have developed a complex network of enzymatic antioxidant systems for handling reactive oxygen species (ROS) generated during stress. Superoxide dismutases (SODs) play a determinant role in balancing ROS in insect. However, studies devoted to SODs functions in insects under cold stress are limited. In the present study, we attempted to identify and characterize a mitochondrial manganese SOD (mMn-SOD) from the desert beetle *Micordera punctipennis* (denoted as MpmMn-SOD) and explore its protective effects on bacteria cells under cold stress. MpmMn-SOD is composed of 202 amino acids with conserved domains required for metal ions binding and enzyme activity. RT-qPCR experiments revealed that the expression of *MpmMn-SOD* was ubiquitous but tissue-specific and was induced by cold stress. An *E. coli* (BL21) system was applied to study the function of MpmMn-SOD. The MpmMn-SOD gene was cloned into the prokaryotic expression vector pET-32a to generate a recombinant plasmid pET-32a(*MpmMn-SOD*). After transformation of the plasmid into *E. coli* BL21, the fusion protein Trx-His-MpmMn-SOD was overexpressed and identified by SDS-PAGE and Western blotting. Antioxidant activity assay showed that the death zones of the transformed bacteria BL21 (pET32a-mMn-SOD) were smaller in diameter than the control bacteria BL21 (pET32a). Survival curves under −4 °C showed that BL21 (pET32a-mMn-SOD) had significant enhanced cold resistance compared to BL21 (pET32a). Its SOD activity under −4 °C had a significant negative correlation ($r = -0.995$) with superoxide anion $O_2^{\bullet-}$ content. Accordingly, under cold stress BL21 (pET32a-mMn-SOD) had lower electric conductivity and malondialdehyde (MDA) content than BL21 (pET32a). Taken together, our results showed that cold stress stimulated the expression of *MpmMn-SOD* in *M. punctipennis*. The *E. coli* cells that overexpress MpmMn-SOD increase their resistance to cold stress by scavenging ROS, and mitigate potential cell damage caused by ROS under cold conditions.

## INTRODUCTION

Oxygen is essential for most life forms. The full reduction of oxygen to $H_2O$ by cytochrome oxidase is a key step in the mechanism of aerobic ATP formation

Corresponding author
Xiaoning Liu, liuxn0103@sina.com

(*Hermes-Lima, Storey & Storey, 2001*). However, the partial reduction of oxygen leads to the formation of various reactive oxygen species (ROS). Superoxide anion radical ($O_2^{\bullet-}$) is usually the first ROS to be generated. The equilibrium between the production and the scavenging of ROS may be perturbed by various biotic and abiotic stress factors such as salinity, UV radiation, drought, heavy metals and temperature extremes (*Sarvajeet Singh & Narendra, 2010*). Insects are constantly subjected to changes in environmental temperature. Low temperature is a major environmental constraint that impacts the geographic distribution and seasonal activity patterns of insects (*Denlinger, 2010*). Cold stress may result in oxidative stress with the accumulation of ROS (*Gharari et al., 2014*; *Jithesh et al., 2006*).

Unbalanced high levels of ROS in living organisms under stress can cause potential damage to biological macromolecules (*Jaramillo-Gutierrez et al., 2010*). To defend against the oxidative injury of ROS, cells are equipped with myriad antioxidant enzymes to scavenge and detoxify the accumulated oxyradicals (*Arenas-Ríos et al., 2007*; *Park, Yang & Yoo, 2004*; *Vaughan, 1997*). Enhanced antioxidants could provide this same action to support winter survival by cold-hardy insects (*Denlinger, 2010*). $O_2^{\bullet-}$, the dominant ROS, is converted to hydrogen peroxide ($H_2O_2$) by superoxide dismutase (SOD), then transformed to water via catalase (CAT) or glutathione peroxidase (GPx) (*Felton & Summers, 1995*; *Schafer & Buettner, 2001*).

SODs are the main antioxidant enzyme families in organisms. They are considered as the first defense line against oxidative stresses due to their functions of converting $O_2^{\bullet-}$ to $H_2O_2$ and $H_2O$ (*Ackerman & Brinkley, 1966*; *McCord & Fridovich, 1988*). SOD is unique in that its activity determines the contents of $O_2$ and $H_2O_2$, the two Haber-Weiss reaction substrates, and it is therefore, central in the defense mechanism (*Bowler, 1992*). SODs are classified into three distinct groups in eukaryotes: intracellular copper/zinc SOD (icCuZn-SOD), extracellular copper/zinc SOD (ecCuZn-SOD) and manganese SOD (Mn-SOD) (*Zelko, Mariani & Folz, 2002*).

Mn-SOD has received much attention because mitochondria is the main source of ROS (*Kailasam et al., 2011*; *Li et al., 2011*). Two types of Mn-SOD are known in eukaryotes: mitochondrial Mn-SOD (mMn-SOD) that has a mitochondrial transit peptide for translocation and cytosolic Mn-SOD (cMn-SOD) without the peptide (*Lin et al., 2010*). Temperature stress was reported as one of the key mediators for ROS generation (*Harari, Fuller & Gerner, 1989*; *Rauen et al., 1999*). The mitochondrial electron-transport chain is responsible for a significant proportion of intracellular superoxide radical production (*Møller, 2010*). Low temperature can fall down the rate of enzymatic reactions, leading to a decrease in demand for ATP and accumulation of electrons in certain points of the respiratory chain. This situation promotes a sudden increase in the production of several ROS which relieve the burden of excess reducing potential. Cold stress is therefore associated with an increased intracellular oxidative stress, and an increase in antioxidants activity appears to be one of features of cold-adaptations (*Chattopadhyay, 2002*). Previous studies have shown that the expression of *Mn-SOD* gene is induced in response to cold stress in several insect species (*Kim et al., 2010*; *Gao et al., 2013b*; *Gao et al., 2013a*; *Jia et al., 2014*). Beetle *Micordera punctipennis* (Coleoptera: Tenebrionidae) is an endemic

species in Gurbantunggut Desert in Xinjiang, China (*Huang et al., 2005*). The adult is cold hardy, the average temperatures of the soil surface and soil-in-5 cm in January were $-12\ ^\circ C$ and $-5\ ^\circ C$, respectively (*Hou et al., 2010*). In the low temperature transcriptome of *M. punctipennis*, GO (Gene ontology) term analysis showed that Mn-SOD is one of the eight significantly up-regulated genes that are related to abiotic stress response (*Tusong et al., 2016*). It is possible that Mn-SOD might be present in the mitochondrial matrix, near the primary source of superoxide, as occurs in other species and may respond to oxidative stress caused by cold stress. However, the function and characteristics of this protein in *M. punctipennis* are currently unclear. In this study we aim at (1) isolating and characterizing a mitochondrial Mn-SOD gene (*MpmMn-SOD*) from *M. punctipennis*; (2) investigating *MpmMn-SOD* distribution patterns in different tissues and temporal expression profiles at overall mRNA levels after being challenged by low temperature in order to explore one of the possible mechanisms of the insect response to cold stress; (3) analyzing the antioxidant activity of the recombinant MpmMn-SOD and the $O_2^{\bullet-}$ content under cold stress by over-expressing this protein in bacteria; (4) examining the protective effects of MpmMn-SOD son the bacteria cells carrying *MpmMn-SOD* gene under cold stress. The results will help to primarily study the possible function of MpmMn-SOD in the desert beetle under cold conditions.

## MATERIALS & METHODS

### Insect treatments, total RNA extraction and cDNA synthesis

The beetles were collected from the wild field in Wujiaqv (N44°29′, E87°31′, 410 m), which is about 100 km northeast of the geological center of Asia. The samples were returned to the laboratory and kept in large plastic beakers containing dry sands at $30 \pm 0.5\ ^\circ C$, 16:8 h (light: dark) photoperiod and relative humidity (RH) of $30 \pm 6\%$. Adults were fed with wheat bran and fresh cabbage leaves.

Beetles were dissected in cold $1 \times$PBS (phosphate belanced solution) to isolate different tissues, such as head, midgut, hindgut (containing Malpighian tubule), fat body and carcass (whole body after head, gut and fat body were removed). The dissected tissues were immediately frozen in liquid nitrogen for RNA extraction.

As $4\ ^\circ C$ is the low temperature at which the insect begins to respond to cold stress (*Hou et al., 2010*), the beetle individuals were exposed at $4\ ^\circ C$ for different time periods (0.5 h, 1 h, 1.5 h, 2 h, 3 h, 5 h, 7 h, 9 h and 11 h, respectively, three replicates per treatment group). The individuals at room temperature (about $25\ ^\circ C$) without any cold treatment were used as control (0 h). After the cold treatment, beetles were immediately frozen in liquid nitrogen for RNA extraction. Total RNA extraction was performed by using Trizol reagent (Invitrogen, Carlsbad, CA) following the manufacturer's protocol. RNA concentration was quantified by using a Nano-Drop ND-1000 spectrophotometer (NanoDrop Technologies, Wilmington, USA). The cDNA was synthesized from 1.0 μg total RNA based on Reverse Transcriptase M-MLV (Takara, China) according to the manufacturer's instructions.

**Table 1  Primer sequences used in this study.**

| Primer name | Primer sequence (5′→3′) | Description |
|---|---|---|
| Primers for RACE | | |
| Mp MnSOD-F1 | TCGAAGTGTTGTTGGACGGGCTATC | 3′-RACE first round PCR |
| Mp MnSOD-F2 | CCCTAGCTTGTGGCGTTATCGCTTA | 3′-RACE second round PCR |
| Universal Primer A Mix (UPM) | CTAATACGACTCACTATAGGGCAAGC | |
| | AGTGGTATCAACGCAGAGT | 3′, 5-′ RACE first round PCR |
| Universal Primer Short | AAGCAGTGGTATCAACGCAGAGT | 3′, 5-′ RACE second round PCR |
| Primers for qRT-PCR | | |
| Mp MnSOD-RT-F | CGCATTTCAACCCTTACCTGT | qRT-PCR forward primer |
| Mp MnSOD-RT-R | ATAGCCCGTCCAACAACACT | qRT-PCR reversed primer |
| Primers for ORF amplification | | |
| Mp MnSOD-ORF-F | GCGGATCCATGTTAACGGTGCTAGCGCTGTGCG | ORF forward primer |
| Mp MnSOD-ORF-R | CCGCTCGAGCGGTTACGTAAGCGATAACGCCACAAG | ORF reversed primer |

## Cloning of the full-length *MpmMn-SOD* cDNA

*MpmMn-SOD* fragment (transcriptomic ID c41919) which had the up-regulated expression at 4 °C was selected from the transcriptomic data of *M. punctipennis* (*Tusong et al., 2017*). The lacked 3′-sequence were obtained by SMARTer^TM RACE cDNA Amplification Kit (Clontech, Beijing, China). Primers used in this experiment were detailed in Table 1. The PCR program was 95 °C for 5 min followed by 30 cycles of 94 °C for 30 s, 63 °C for 30 s, 72 °C for 1 min and a final extension at 72 °C for 10 min. For verification, PCR products were purified, and cloned into pMD18-T vector (Trans GenBiotech, Beijing, China), and then were transformed into competent *E. coli* cells (DH5a) for Sanger sequencing by Quintarabio, Urumqi, China.

The deduced amino acid domains in MpmMn-SOD were analyzed using the BLAST search program (http://blast.ncbi.nlm.nih.gov/Blast.cgi). The physicochemical properties were predicted by using AntPASy's ProtParam Online Tool. Multiple sequence alignments among insect species in different orders were created with DNAMAN 6.0 software (http://www.lynnon.com). The signal peptide cleavage site was examined with SignalP 4.1 (http://www.cbs.dtu.dk/services/SignalP/) program. TargetP 1.1 (http://www.cbs.dtu.dk/services/TargetP/) was used to predict presence of a putative mitochondrial targeting sequence (MTS). Phylogenetic analysis was performed by IQTREE 1.6.2. The phylogenetic tree was constructed based on predicted amino acid sequences using the Maximum Likelihood (ML) method with 5,000 replicates bootstrap. Mn-SOD sequences in different insect species were downloaded from the database in NCBI website.

## Detection of the mRNA level of *MpmMn-SOD* by Fluorescent real-time quantitative PCR (RT-qPCR)

The expression of *MpmMn-SOD* transcript was assayed on a 7500 Real Time PCR System (Applied Biosystems, USA) using SYBR Green Mix to determine the expression profiles of *MpmMn-SOD* gene in different tissues and in the beetle at 4 °C for 0∼11 h as described above. Translation elongation factor (*EF*-α) was used as a reference gene to normalize
the target gene expression levels among samples (*Xikeranmu et al., 2019*). Primers for RT-qPCR are detailed in Table 1. The qPCR amplification conditions were 94 °C for 2 min, followed by 40 cycles at 94 °C for 30 s and 62 °C for 30 s. The relative expression of the target gene was calculated using the comparative $2^{-\triangle\triangle CT}$ method. The change of the gene expression levels at 4 °C was normalized to the gene in the control (0 h). The value at each time point was given as mean $\pm$ *S.E.* ($n = 3$).

The expression of *MpmMn-SOD* mRNA in head, midgut, hindgut, fat body and carcass were separately detected by RT-qPCR. The mRNA levels of *MpmMn-SOD* in different tissues were normalized to that of the head which had the lowest expression level. The value was given as mean $\pm$ *S.E.* ($n = 3$).

## Prokaryotic expression and Western blot analysis of the fusion protein Trx-His-MpmMn-SOD

To obtain the recombinant MpmMn-SOD protein and examine whether it possesses antioxidant activity, DNAMAN was used to design primers containing *Bam* HI and *Xho* I restriction sites (Table 1) to amplify the coding sequence (CDS) of this gene. The amplified fragments were digested with the endonucleases, and subcloned into a pET-32a (+) expression vector that was digested with the same enzymes. The constructed plasmid denoted as pET-32a (*MpmMn-SOD*) was transformed into competent cells of *E. coli* BL21 (DE3). The parent vector pET-32a without inserts gene was transformed into BL21 (DE3), and used as a control. The two transformed bacteria, BL21(pET-32a-MpmMn-SOD) and BL21(pET-32a), were induced with 0.5 mM isopropyl β-D-thiogalactoside (IPTG) at 25 °C for 10 h to overexpress fusion proteins Trx-His-MpmMn-SOD (41 kDa) and Trx-His (the tag protein on the vector, 18.5 kDa) respectively in *E. coli*. LB (Luria-Bertani) broth was used for bacterial culture medium. Expression efficiency of different transformants was assessed by analysis of the target protein band in sodium dodecyl sulfate -polyacrylamide gel electrophoresis (SDS-PAGE). The correct expressions of these proteins were further confirmed by Western blotting with anti-His antibody (Zsbiotech, Beijing, China).

## Antioxidant activity assay by the Oxford cup method

To test whether MpmMn-SOD has antioxidant activity, the tolerance of the Trx-His-MpmMn-SOD-overexpressed *E. coli* cells to hydroperoxide was determined by the Oxford cup method (*Liu et al., 2018*). The bacteria BL21(pET-32a-MpmMn-SOD) and BL21(pET-32a) were grown overnight at 37 °C in LB broth containing ampicillin (Amp$^+$) (50 mg/L), and then diluted 1: 100 in LB medium. The diluted cells were further incubated at 37 °C until a final optical density of 0.4~0.6 at 595 nm. These cells were induced with 0.3 mM IPTG at 25 °C for 10 h. Then, 100 μL BL21(pET-32a-MpmMn-SOD) and BL21(pET-32a) were, respectively, added to fresh LB (Amp$^+$) solid medium in plates. After the medium is solidified, five Oxford cups were placed on the plate, then 100 μL of different concentrations (100, 75, 50, 25 and 0 mmol/L) of $H_2O_2$ were respectively added to the top of the cups. The plates were incubated overnight at 37 °C with three replicates per treatment group. BL21(pET-32a) cells were used as the control. The agent diffused into the surrounding area through the cup to form a decreasing concentration gradient. Observe the zone of

inhibition formed around the cup and record the diameter of the zone. The inhibition zones were measured as described by *Burmeister et al. (2008)*.

## Measurement of SOD activity and $O_2^{\bullet-}$ content in the MpmMn-SOD overexpressed bacteria at $-4\ °C$

As 4 °C is not enough to influence bacteria survival within short time, we treated the MpmMn-SOD-overexpressed bacteria under $-4$ °C. BL21(pET-32a) were set as the control. Cultures of the two bacteria were induced to produce proteins Trx-His-MpmMn-SOD and Trx-His separately by addition of IPTG described above, and five mL cultures of the bacteria were exposed to $-4$ °C for 0 h, 2 h, 4 h and 6 h, respectively. At the end of the cold treatments, the bacteria in each group were recovered at 37 °C for 1 h, and OD595 was determined for making survival curve. The control was $-4$ °C for 0 h no cold treatment. Each treatment had three replicates.

Then, the cells were harvested by centrifugation (12,000 rmp,10 min, at 4 °C). The collected cells were sonicated in an ice bath after suspension in PBS. The supernatants were collected as crude enzyme liquids and were quantified using the BCA Protein Assay Kit (Thermo Scientific Pierce, IL, USA). $O_2^{\bullet-}$ content was measured according to the hydroxylamine oxidation method described by *Wang & Luo (1990)*. The SOD activity was determined using the CuZn/Mn-SOD Assay Kit (Jiancheng, Nanjing, China) following the manufacturer's protocol with minor modification (*Meng et al., 2013*).

The relative increase of SOD activity was calculated by subtracting the SOD activity of BL21(pET-32a-MpmMn-SOD) from that of the control BL21(pET-32a), then dividing the difference by the SOD activity of BL21(pET-32a-MpmMn-SOD), so did for calculating the relative decrease of $O_2^{\bullet-}$ content.

## Measurement of relative electrical conductivity (REC) and malondialdehyde (MDA) content of bacteria BL21 (pET-32a-MpmMn-SOD) under $-4\ °C$

The influence of low temperature on cell membrane permeability was determined by measuring the relative electrical conductivity (REC) in bacteria BL21(pET-32a-MpmMn-SOD) and BL21(pET-32a) respectively. BL21(pET-32a) was used as the control. After the recombinant protein was over-expressed by IPTG induction, five mL of the bacteria were centrifuged at 6,000 rpm for 10 min, then the cells were collected and washed with 5% dextrose solution until the bacterial solution's REC (denoted as L1) was comparable to that of 5% glucose solution. Then, five mL of the isotonic bacterial solutions was stored at $-4$ °C for 0 h, 2 h, 4 h and 6 h, respectively, following which REC (denoted as L2) was measured for each time point. After boiling for 5 min, the REC (denoted as L0) of the treated bacteria solution was measured again. The final relative conductivity was calculated as: REC (%) = 100 × (L2 -L1)/L0.

For MDA determination, five mL of the cultures were exposed at $-4$ °C for 0 h, 2 h, 4 h and 6 h respectively after IPTG induction. Then, 400 μL MDA extract solution (MDA Assay Kit, Solarbio, Beijing, China) were added to the bacteria cells (about two million) to lyse the cells. The mixture was centrifuged at 8,000 rpm for 10 min at 4 °C. The supernatant

was collected and set on ice bath. The MDA content was determined by using MDA Assay Kit (Solarbio, Beijing, China) according to the manufacturer's protocol.

## Statistical analysis

One-way analysis of variance and Tukey's multiple comparison test were conducted for data analysis for gene expression, SOD activity measurement and $O_2 \bullet^-$ content determination. Paired $t$-test was employed for analyzing data from experiments measuring diameter of death zone, relative conductivity and MDA content. Spearman's correlation analysis was used for correlation analysis of SOD activity and $O_2 \bullet^-$ content. Data were shown as mean $\pm$ *S.E.*

## Commonly used acronyms

ecCuZn-SOD: extracellular copper/zinc superoxide dismutase; GO: gene ontology; icCu-Zn/SOD: intracellular copper/zinc superoxide dismutase; IPTG: Isopropyl β-D-thiogalactoside; Mn-SOD: manganese SOD; mMn-SOD: mitochondrial Mn-SOD; MTS: mitochondrial targeting sequence; MDA: malondialdehyde; PBS: phosphate balanced solution; REC: relative electrical conductivity; SDS-PAGE: sodium dodecyl sulfate-polyacrylamide gel electrophoresis; Trx-His: thioredoxin–histone.

# RESULTS

## Identification and characterization of the MpmMn-SOD sequence

We obtained the *MpmMn-SOD* sequence from the transcriptomic data of *M. punctipennis*, and then confirmed this sequence by cDNA cloning. The full-length cDNA was 1,359 bp including an open reading frame (ORF) of 609 bp, a 3′-UTR of 750 bp with a poly (A) tail and a single polyadenylation signal (AATAAA). The ORF encoded a protein of 202 amino acids (GenBank accession no. MK676072.1), the calculated molecular mass was 22 kDa, and the estimated pI was 6.54, no signal peptide was predicted. The protein was predicted as a Mn-SOD containing one N-glycosylation site (NGTL) and a putative N-terminal mitochondrial targeting sequence (MTS) which was consisted of 12 amino acids (Fig. 1), suggesting this protein may exist in mitochondria. We designated this sequence MpmMn-SOD.

Comparison of the predicted amino acids of MpmMn-SOD with Mn-SODs from different insect species revealed the high conservation of four manganese binding sites (His26, His76, Asp162 and His166). One signature of Mn-SOD from 162 to 169 (DV/IWEHAYY) was also conserved across these insect species (Fig. 2). MpmMn-SOD was most like the yellow meal worm Mn-SOD, the two sequences both had a shortened N-terminal sequence compared to Mn-SODs from insects in other taxonomic order. However, the identity of these two sequences was only 35.27%, suggesting that MpmMn-SOD was a novel insect Mn-SOD.

To further analyze MpmMn-SOD sequence with SOD sequences in other insects at evolutionary perspective, phylogenetic analysis was conducted. The results revealed two separate clusters, Mn-SOD and Cu/Zn-SOD, in the phylogenetic tree with strong bootstrap (100%) support, in accordance with their distinct metal cofactor requirements (Fig. 3).

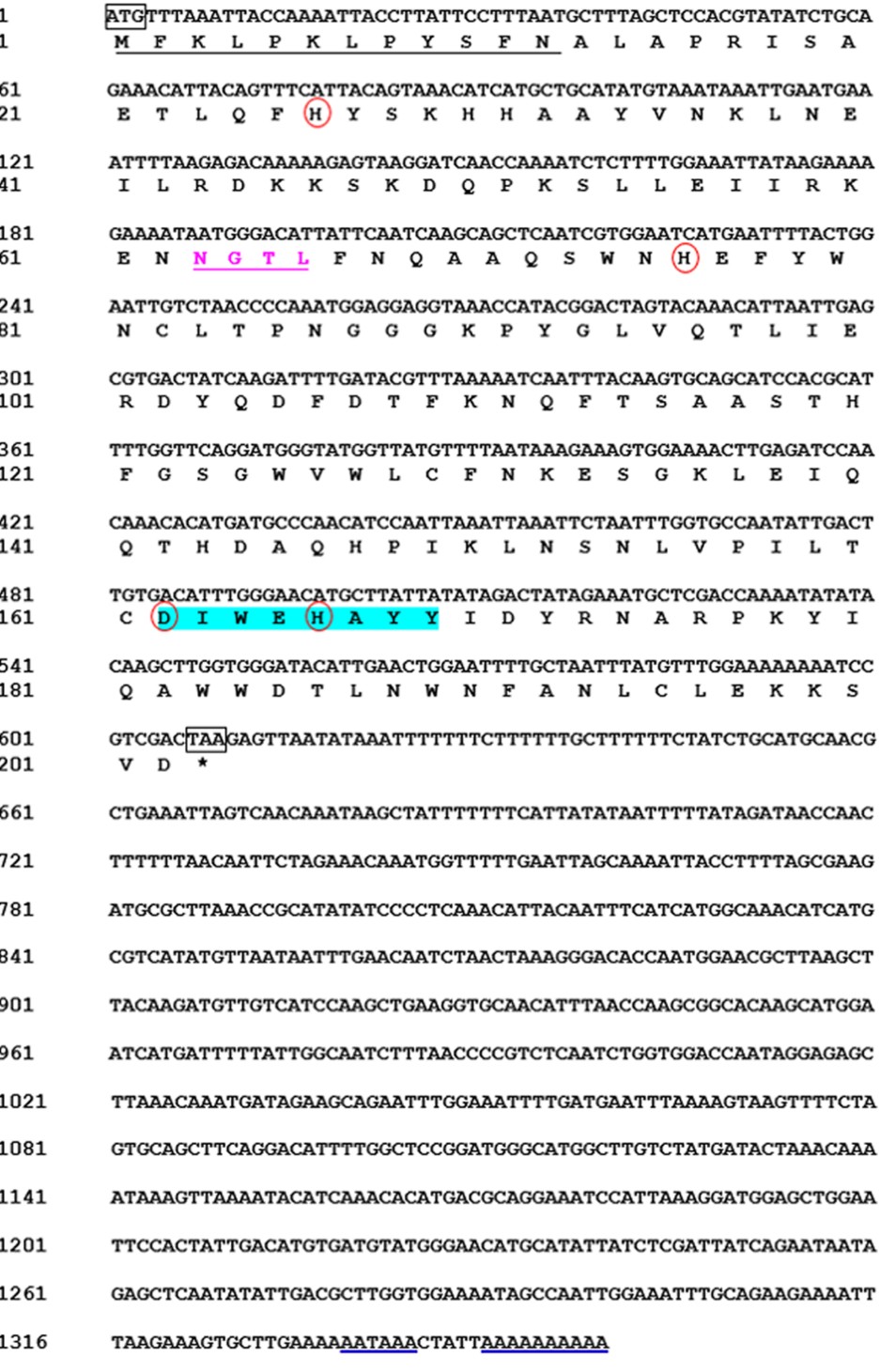

```
1     ATG TTTAAATTACCAAAATTACCTTATTCCTTTAATGCTTTAGCTCCACGTATATCTGCA
1     M  F  K  L  P  K  L  P  Y  S  F  N  A  L  A  P  R  I  S  A

61    GAAACATTACAGTTTCATTACAGTAAACATCATGCTGCATATGTAAATAAATTGAATGAA
21    E  T  L  Q  F (H) Y  S  K  H  H  A  A  Y  V  N  K  L  N  E

121   ATTTTAAGAGACAAAAAGAGTAAGGATCAACCAAAATCTCTTTTGGAAATTATAAGAAAA
41    I  L  R  D  K  K  S  K  D  Q  P  K  S  L  L  E  I  I  R  K

181   GAAAATAATGGGACATTATTCAATCAAGCAGCTCAATCGTGGAATCATGAATTTTACTGG
61    E  N  N  G  T  L  F  N  Q  A  A  Q  S  W  N (H) E  F  Y  W

241   AATTGTCTAACCCCAAATGGAGGAGGTAAACCATACGGACTAGTACAAACATTAATTGAG
81    N  C  L  T  P  N  G  G  G  K  P  Y  G  L  V  Q  T  L  I  E

301   CGTGACTATCAAGATTTTGATACGTTTAAAAATCAATTTACAAGTGCAGCATCCACGCAT
101   R  D  Y  Q  D  F  D  T  F  K  N  Q  F  T  S  A  A  S  T  H

361   TTTGGTTCAGGATGGGTATGGTTATGTTTTAATAAAGAAAGTGGAAAACTTGAGATCCAA
121   F  G  S  G  W  V  W  L  C  F  N  K  E  S  G  K  L  E  I  Q

421   CAAACACATGATGCCCAACATCCAATTAAATTAAATTCTAATTGGTGCCAATATTGACT
141   Q  T  H  D  A  Q  H  P  I  K  L  N  S  N  L  V  P  I  L  T

481   TGTGACATTTGGGAACATGCTTATTATATAGACTATAGAAATGCTCGACCAAAATATATA
161   C  D  I  W  E  H  A  Y  Y  I  D  Y  R  N  A  R  P  K  Y  I

541   CAAGCTTGGTGGGATACATTGAACTGGAATTTTGCTAATTTATGTTTGGAAAAAAAATCC
181   Q  A  W  W  D  T  L  N  W  N  F  A  N  L  C  L  E  K  K  S

601   GTCGAC TAA GAGTTAATATAAATTTTTTTCTTTTTTGCTTTTTTCTATCTGCATGCAACG
201   V  D  *

661   CTGAAATTAGTCAACAAATAAGCTATTTTTTTTCATTATATAATTTTTATAGATAACCAAC

721   TTTTTTAACAATTCTAGAAACAAATGGTTTTTGAATTAGCAAAATTACCTTTTAGCGAAG

781   ATGCGCTTAAACCGCATATATCCCCTCAAACATTACAATTTCATCATGGCAAACATCATG

841   CGTCATATGTTAATAATTTGAACAATCTAACTAAAGGGACACCAATGGAACGCTTAAGCT

901   TACAAGATGTTGTCATCCAAGCTGAAGGTGCAACATTTAACCAAGCGGCACAAGCATGGA

961   ATCATGATTTTTATTGGCAATCTTTAACCCCGTCTCAATCTGGTGGACCAATAGGAGAGC

1021  TTAAACAAATGATAGAAGCAGAATTTGGAAATTTTGATGAATTTAAAAGTAAGTTTTCTA

1081  GTGCAGCTTCAGGACATTTTGGCTCCGGATGGGCATGGCTTGTCTATGATACTAAACAAA

1141  ATAAAGTTAAAATACATCAAACACATGACGCAGGAAATCCATTAAAGGATGGAGCTGGAA

1201  TTCCACTATTGACATGTGATGTATGGGAACATGCATATTATCTCGATTATCAGAATAATA

1261  GAGCTCAATATATTGACGCTTGGTGGAAAATAGCCAATTGGAAATTTGCAGAAGAAATT

1316  TAAGAAAGTGCTTGAAAAAATAAACTATTAAAAAAAAAA
```

**Figure 1** **Nucleotide and deduced amino acid sequences of a Mn-SOD from the beetle *M. punctipennis*.** The start codon (ATG) and the stop codon (TAA) are boxed. The putative mitochondrial targeting sequence (MTS) is underlined in black. The potential N-glycosylation site (NGTL) is shown and underlined in pink. The Mn-SOD signature motif (DIWEHAYY) is highlighted in cyan. The putative polyadenylation signal (AATAAA) and poly (A) are underlined in blue. The four highly conserved amino acids (His26, His76, Asp162, and His166) critical for Mn-binding are circled in red.

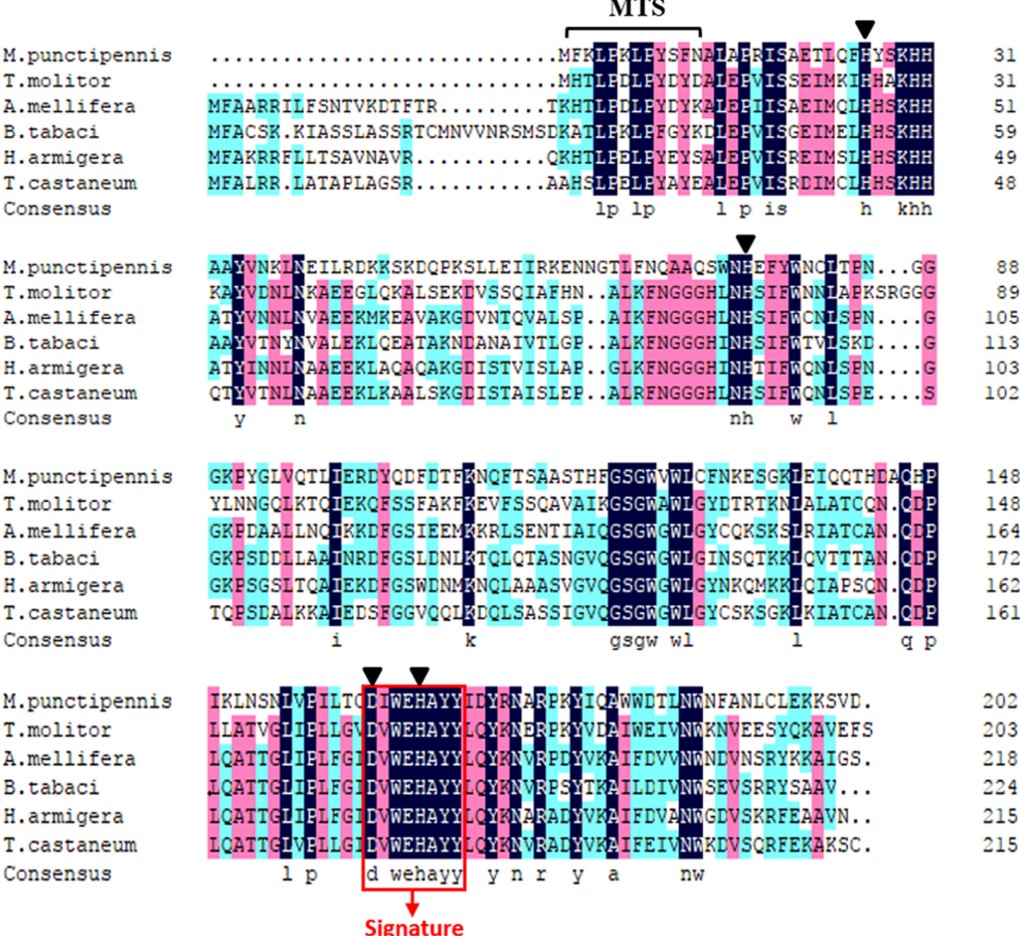

**Figure 2 Multiple alignments of the deduced amino acid sequences of the Mn-SODs from the beetle *M. punctipennis* and other known insect species.** The Mn-SOD signature DIWEHAYY is boxed in red (labeled Signature). Mn-binding sites are indicated with triangles. The amino acid homology up to 100% is shown in black, 75% ∼99% is in pink and 50% ∼74% in cyan. *M. punctipennis*: *Microdera punctipennis*; *T. molitor*: *Tenebrio molitor*; *A. mellifera*: *Apis mellifera*; *B. tabaci*: *Bemisia tabaci*; *H. armigera*: *Helicoverpa armigera*; *T. castaneum*: *Tribolium castaneum*.

MpmMn-SOD was clustered with Mn-SODs. Within Cu/Zn-SOD clade, ecCu/Zn-SOD and icCu/Zn-SOD were classified as two subgroups with strong bootstrap support (98%), and ecCu/Zn-SOD subgroup was the basic form. MpmMnSOD was closed to the Mn-SOD from the yellow meal worm *Tenebrio molitor*.

## Expression of the *MpmMn-SOD* gene in different tissues

To examine the tissue distribution profile of *MpmMn-SOD* expression, the mRNA levels from head, midgut, hindgut, fat body and carcass were measured by using RT-qPCR. The results showed that *MpmMn-SOD* expressed in all these tissues, but the expression levels varied greatly among these tissues. The highest was in hindgut followed by fat body, midgut and carcass; the lowest was in head (Fig. 4A). The expression levels in hindgut, fat body,

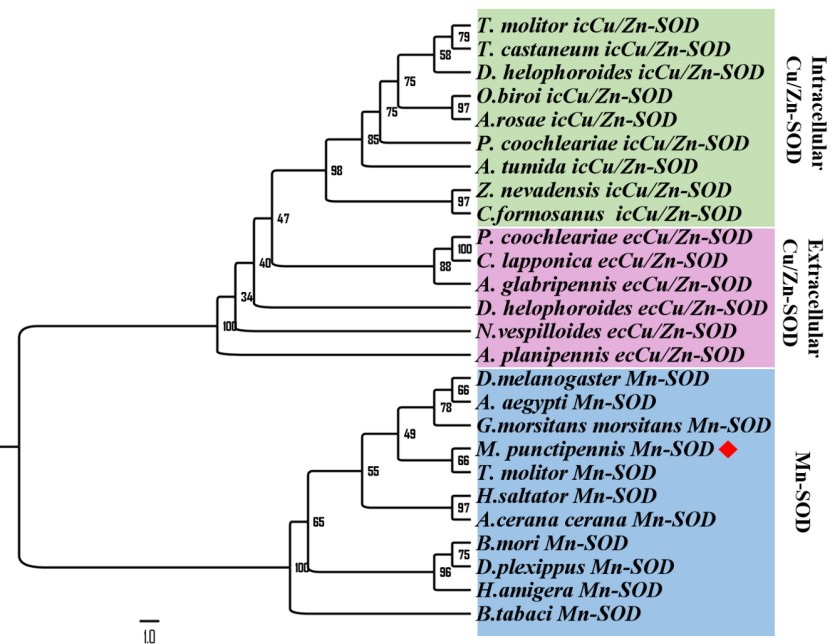

**Figure 3** **Phylogenetic analysis of SOD sequences from *M. punctipennis* and other insect species based on predicted amino acid sequences.** *A. cerana cerana: Apis cerana cerana; A. glabripennis: Anoplophora glabripennis; A. planipennis: Agrilus planipennis; A. rosae: Athalia rosae; A. tumida: Aethina tumida; B. mori: Bombyx mori; B. tabaci: Bemisia tabaci; C. formosanus: Coptotermes formosanus; C. lapponica: Chrysomela lapponica; D. helophoroides: Dastarcus helophoroides; D. melanogaster: Drosophila melanogaster; D. plexippus: Danaus plexippus; G. morsitans morsitans Glossina morsitans morsitans; H. saltator: Harpegnathos saltator; M. punctipennis: Microdera punctipennis; O. biroi: Ooceraea biroi; P. coochleariae: Phaedon cochleariae; T. castaneum: Tribolium castaneum; T. molitor: Tenebrio molitor; Z. nevadensis: Zootermopsis nevadensis.*

midgut and carcass were 57-fold, 17-fold, 5.3-fold and 3.5-fold of the head, respectively ($F_{(4,10)} = 111.645$, $P < 0.01$), suggesting a tissue specific expression pattern.

## Temporal expression of *MpmMn-SOD* in *M. punctpennis* at 4 °C

To test whether *MpmMn-SOD* expression is cold inducible and how does cold affect *MpmMn-SOD* expression in the insect, the overall mRNA expression profile at 4 °C for different time periods was determined by RT-qPCR. As 4 °C is the low temperature at which the insect begins to respond to cold stress, we exposed the beetles to 4 °C for 0 h ~11 h. The results showed that the mRNA level of *MpmMn-SOD* was significantly increased after the cold exposure compared with the control (0 h, no cold treatment), and this simulative effect was very significant ($F_{(9,18)} = 80.07$, $P < 0.001$) (Fig. 4B). It was approximately 9.9-fold, 22.5-fold and 125-fold of the control after cold exposure for 0.5 h,1 h and 1.5 h ($P < 0.01$) respectively. The second large expression peak appeared at 11 h, which was about 67.3-fold of the control (0 h). From 2 h to 9 h the levels slightly fluctuated with a small peak of 16-fold of the control at 3 h. This cold expression profile presented a stress-responsive pattern. The large fluctuation of *MpmMn-SOD* expression during the cold treatment indicated that cells could adjust the level of the enzyme timely and precisely.

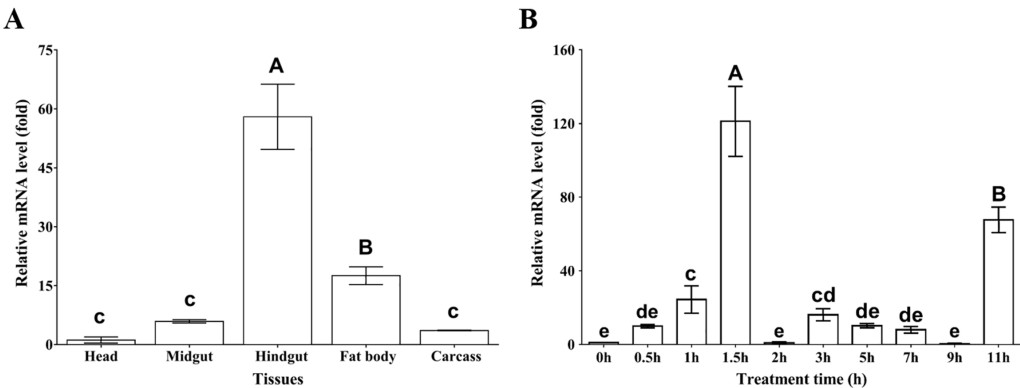

**Figure 4** **Relative mRNA levels of *MpmMn-SOD* detected by RT-qPCR.** (A) Expression profile of *MpmMn-SOD* in different tissues. Value is fold compared to the head; (B) Temporal expression of *MpmMn-SOD* gene in the beetle under 4 °C cold stress. Value is fold compared to the control (0 h). Different letters on the top of each bar indicate statistical significance. $P < 0.05$ (lower-case letters), $P < 0.01$ (capital letters). In Fig. 4A bars of hindgut and fat body labelled with A and B respectively indicate very significant difference ($P < 0.01$) between the two tissues and between them and the rest tissues; bars labelled with c indicate no significant differences among them ($P > 0.05$). In B, bars of 1.5 h and 11 h labelled with A and B respectively indicate very significant difference ($P < 0.01$) between the two time points and between them and the rest time points; bars of 1 h and 3 h labelled with c and cd respectively indicate no significant difference ($P > 0.05$) between them, but they have significant differences ($P < 0.05$) with bars labelling e; bars labelled with cd and de indicate no significant difference ($P > 0.05$) among them; bars labelled with e and de indicate no significant difference ($P > 0.05$) among them.

## Prokaryotic expression and Western blot analysis of the fusion protein Trx-His-MpmMn-SOD

To study the enzyme activity of MpmMn-SOD, we inserted *MpmMn-SOD* into pET32a expression vector and transformed *E. coli* BL21 with the recombinant plasmid. pET32a alone was also transformed into *E. coli* BL21 as the control. The fusion protein Trx-His-MpmMn-SOD and the tag protein Trx-His were separately over-expressed in the two transformants through IPTG induction. The expression of these two proteins were analyzed on SDS-PAGE (Fig. 5A). A clear thick band of about 41 kDa appeared in lane 4 after IPTG induction, it matched the calculated size of the molecular mass of Trx-His-MpmMn-SOD; and a clear thick band of about18.5 kDa appeared in lane 2 after IPTG induction, it matched the calculated size of the molecular mass of Trx-His. The two proteins were absent in the un-induced samples. We further confirmed the two fusion proteins by Western blotting using anti-His antibody (Fig. 5B). The results indicated that Trx-His-MpmMn-SOD and Trx-His both were correctly expressed in their host *E. coli* cells.

## Oxidative tolerance of the MpmMn-SOD overexpressed BL21 (pET32a-mMn-SOD)

Antioxidant activity assay was performed to evaluate whether the overexpress mMpMn-SOD could enhance the tolerance of BL21 (pET32a-mMn-SOD) to oxidative stress. The results showed that the death zones of each concentration on the BL21 (pET32a-mMn-SOD) plate were smaller than those on the control BL21 (pET32a) plate (Fig. 6A). The

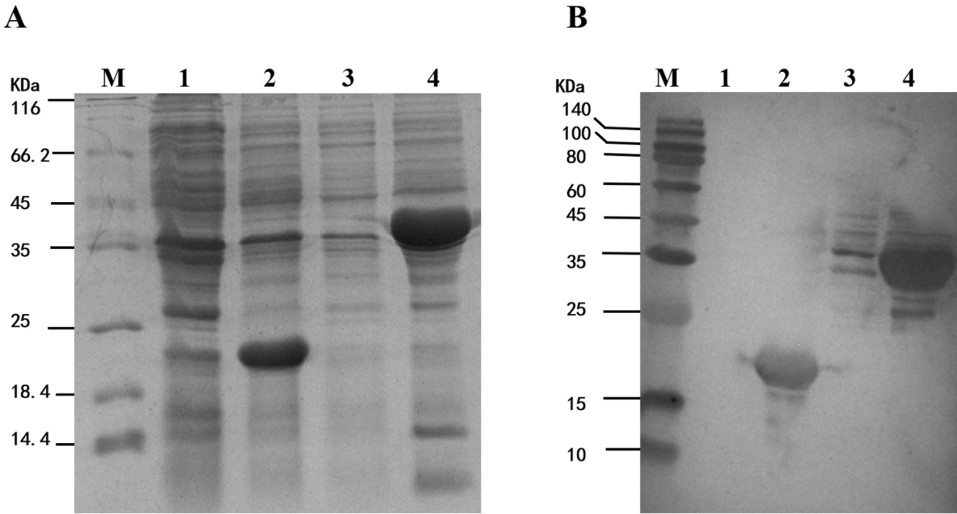

**Figure 5** **Analysis of the fusion protein Trx-His-mMpMn-SOD overexpressed in BL21 cells.** (A) SDS-PAGE analysis of the whole cell lysate. (B) Western blot analysis. M: protein marker; lane1: non-induced BL21(pET32a); lane2: IPTG induced BL21(pET32a); lane3: non-induced BL21(pET32a-mMn-SOD); lane4: IPTG induced BL21 (pET32a-mMn-SOD).

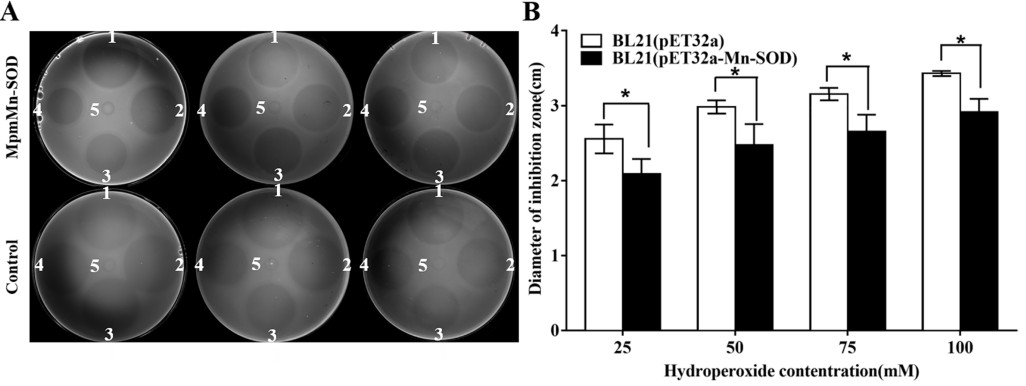

**Figure 6** **Antioxidant activity assay on LB agar plates containing *E. coli* cells overexpressing MpmMn-SOD.** Oxford cups containing different concentrations of $H_2O_2$ were used to generate oxidative stress to the cells. (A) Inhibited zones. Numbers 1~5 on each plate represent $H_2O_2$ concentrations from 100 mmol/L to 0 mmol/L. mMn-SOD: BL21(pET32a-mMn-SOD); Control: BL21(pET32a). (B) Quantitative diameters of the inhibited zones in histograms. Values are compared to the control bacteria in the same group. The data are mean $\pm$ *S.E.* of three replicates.

quantified diameters of the death zones on the BL21 (pET32a-mMn-SOD) plates were significantly smaller than those of the control bacteria (Fig. 6B), demonstrating that the mMn-SOD-overexpressing *E. coli* cells were more tolerant to $H_2O_2$-mediated oxidative damage than the control BL21 (pET32a).

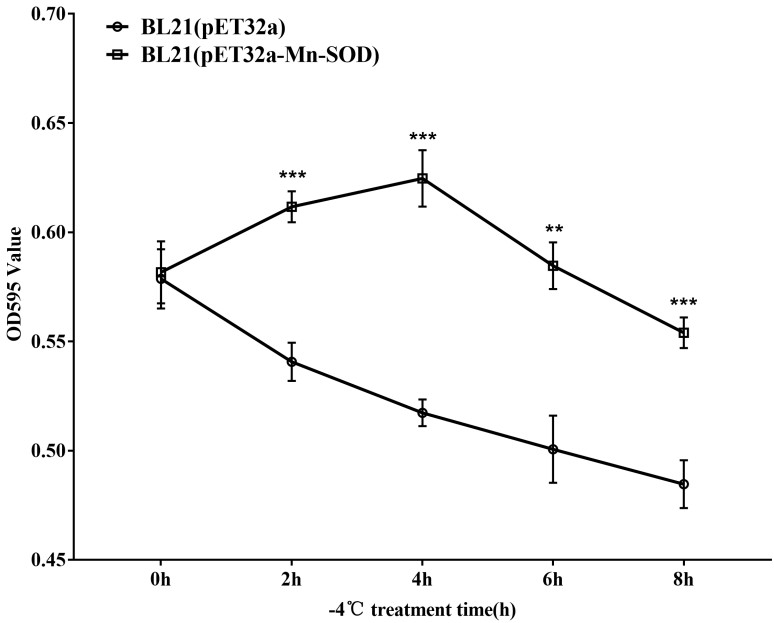

**Figure 7** Survival curve of the MpmMn-SOD overexpressed BL21 (pET32a-mMn-SOD) at −4 °C cold stress.

## Survival curve of the MpmMn-SOD overexpressed BL21 (pET32a-mMn-SOD) under cold stress at −4 °C

As *E. coli* is not significantly harmed by exposure to 4 °C for a limited time, we exposed the MpmMn-SOD overexpressed bacteria BL21(pET32a-mMn-SOD) to −4 °C to test the protective function of MpmMn-SOD for the bacteria under cold stress. After the bacteria were exposed to −4 °C for 0 h, 2 h, 4 h, 6 h and 8 h, respectively, they were recovered at 37 °C for 1 h, and OD595 was determined for making survival curve. The survival curve of BL21(pET32a-mMn-SOD) was a convex type, while it was a rough negative line for the control bacteria (Fig. 7), indicating the cold resistance of BL21(pET32a-mMn-SOD) was significantly increased compared to the control BL21(pET32a). At 4 h of the cold treatment the OD595 of BL21(pET32a-mMn-SOD) was 0.63, significantly higher than the control of 0.52.

## SOD activity and $O_2^{\bullet-}$ content of the MpmMn-SOD overexpressed BL21 (pET32a-mMn-SOD) under −4 °C

We detected the changes of the SOD activity and $O_2^{\bullet-}$ content in the MpmMn-SOD overexpressed bacteria after −4 °C treatment. The results showed that the cold stress significantly stimulated SOD activity of BL21(pET32a-Mn-SOD) compared to BL21(pET32a) (Fig. 8A), suggesting the over-expressed MpmMn-SOD in bacteria not only increased SOD activity overall, but also enhanced the response of SOD activity of the cells to cold stress. Accordingly, the $O_2^{\bullet-}$ content in BL21(pET32a-Mn-SOD) was significantly lower than the control bacteria, suggesting that the over-expressed MpmMn-SOD effectively scavenged $O_2^{\bullet-}$ in cells (Fig. 8B). Pearson's correlation analysis showed

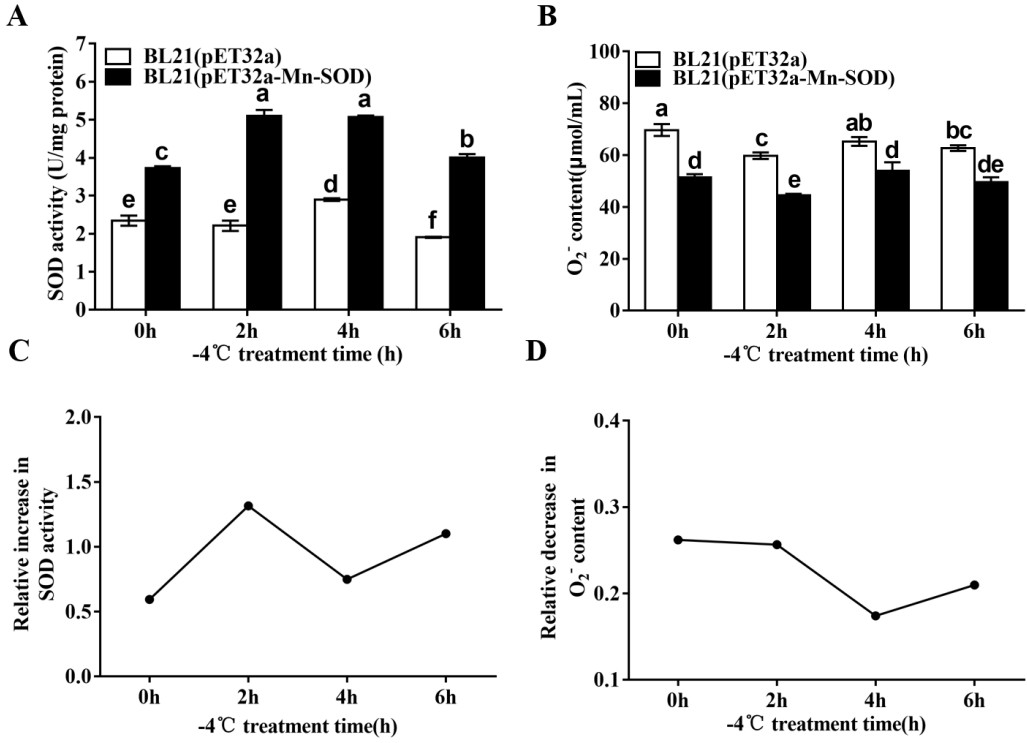

**Figure 8** **SOD activity and $O_2\bullet^-$ content in the MpmMn-SOD overexpressed BL21 (pET32a-mMn-SOD) at $-4\,^{\circ}$C cold stress.** (A) SOD activity. (B) $O_2^{\bullet-}$ content. (C) Relative increase in SOD activity. (D) Relative decrease in $O_2^{\bullet-}$ content. Different letters above each column indicate statistical significance, $P < 0.05$.

that the SOD activity and $O_2\bullet^-$ content of BL21(pET32a-Mn-SOD) under $-4\,^{\circ}$C treatment were strong negatively correlated, and the correlation coefficient was $-0.995$ ($P < 0.05$). The relative increase of SOD activity at 2 h and 6 h of the cold treatment was 2.3-fold and 2-fold of those of the control respectively (Fig. 8C). Correspondingly, at 2 h and 6 h the relative decrease of $O_2\bullet^-$ content (Fig. 8D) was high. The two indexes had similar changing trends, suggesting that the more increase in SOD activity, the more decrease in $O_2\bullet^-$ content under $-4\,^{\circ}$C temperature.

### Relative electrical conductivity (REC) and Malondialdehyde (MDA) content of the MpmMn-SOD overexpressed BL21 (pET32a-mMn-SOD) at $-4\,^{\circ}$C

The excessive accumulation of ROS under cold stress may cause lipid peroxidation which leads to damage of cell membranes. Therefore, electrolyte leakage and MDA level in the BL21 (pET32a-mMn-SOD) and the control BL21 (pET32a) under $-4\,^{\circ}$C were determined. After the cold stress, the two groups of bacteria showed increased REC and MDA content (Fig. 9), indicating that $-4\,^{\circ}$C caused cells damage to both groups. However, the increasing trends of the two indexes in BL21(pET32a-mMn-SOD) were slower than those of the control BL21(pET32a) during the cold treatment. At 4 h and 6 h of the cold treatment, both REC and MDA content in the control group were significantly higher

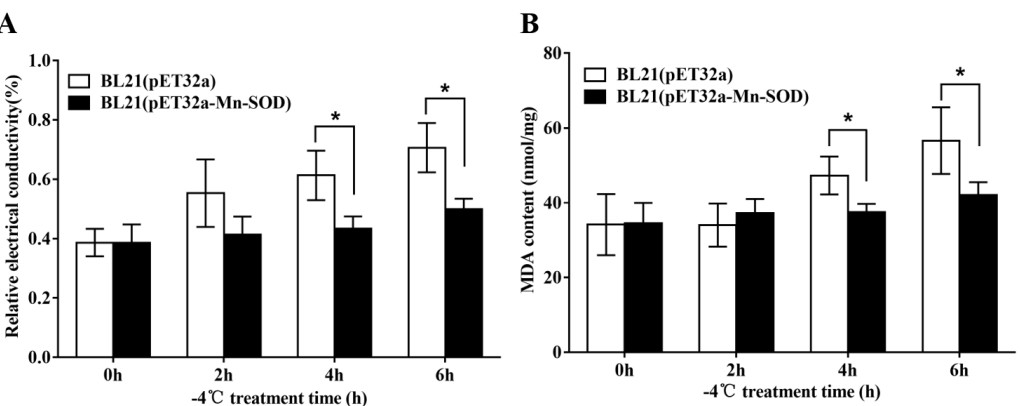

**Figure 9  Protective effect of MpmMn-SOD on bacteria BL21(pET32a-mMn-SOD) at −4 °C.** (A) Relative conductivity. (B) MDA content. Paired *t*-test was conducted to analyze the difference between BL21 (pET32a-mMn-SOD) and BL21 (pET32a) in each treatment group. The symbol * indicates statistical significance $P < 0.05$. Values are mean ± *S.E.* ($n = 3$).

than the experimental group (Figs. 9A, 9B), indicating that the cell membrane injury in BL21(pET32a-mMn-SOD) was less serious than in the control bacteria. These results suggested that the overexpressed MpmMn-SOD conferred cold tolerance to cells via increasing their ability for ROS-scavenging thus reducing membrane damage.

## DISCUSSION

When insects suffer from environmental stresses such as extreme temperatures, reactive oxygen species (ROS) are spawned (*Gao et al., 2013b*; *Gao et al., 2013a*). Metalloenzyme SOD is the most effective intracellular enzymatic antioxidant which is ubiquitous in all aerobic organisms and in all subcellular compartments prone to ROS mediated oxidative stress. It removes highly toxic $O_2 \bullet^-$ and hence prevents the risk of hydroxyl radical OH● generation via the metal catalyzed Haber-Weiss-type reaction (*Fridovich, 1978*). Mn-SOD is considered as a general stress responsive factor whose expression might be influenced by a variety of intracellular and environmental cues including cold stress at transcriptional and/or translational levels (*Cho et al., 2006*; *Zelko, Mariani & Folz, 2002*). Only a little is known to date about oxidative stress induced by cold and functional characterization of SOD in cold-hardy insects. In the present study, a *mMn-SOD* gene, *MpmMn-SOD*, from the desert beetle *Microdera punctipennis* was cloned, characterized and the cold protective effect of MpmMn-SOD protein was investigated for the first time.

Sequence analysis showed that *MpmMn-SOD* encodes four metal-binding residues (His26, His76, Asp162, and His166) and a highly conserved Mn-SOD amino acid motif DI/VWEHAYY, suggesting that these sites were essential to the structure and function of Mn-SODs. The identification of the signature sequence and the conserved metal-binding residues suggested that MpmMn-SOD possessed the essential properties of Mn-SOD family. Many mitochondrial proteins are synthesized as precursors containing MTS (*Yamamoto et*

*al., 2005c*). The finding of MTS in MpmMn-SOD sequence suggested that MpmMn-SOD was of precursor type being transported into the mitochondria.

A BLASTP search at GenBank revealed that MpmMn-SOD sequence was most close to amino acid sequence of Mn-SOD from the yellow meal worm *T. molitor* with identity of 35.27%, indicating that MpmMn-SOD was more diverged from the other Mn-SODs in insects. Phylogenetic analysis confirmed this relationship. These two insects are belonging to the family of Tenebrionidae (Coleoptera), their Mn-SOD sequences both were short at the N-terminal. The close relationship of their Mn-SOD sequences may roughly reflect their taxonomic relationships. The phylogenetic tree revealed that Mn-SOD and Cu/Zn-SOD may originate from a same ancestor, and Mn-SOD may have evolved longer than Cu/Zn-SOD. Besides, Cu/Zn-SOD clade was subdivided into ecCu/Zn-SOD and icCu/Zn-SOD, and ecCu/Zn-SOD subgroup showed more divergency than icCu/Zn-SOD, these two protein sub-families may evolve by gene replication.

Previous study suggested that *Mn-SOD* in insect is widely distributed in a variety of cells and tissues (*Zelko, Mariani & Folz, 2002*). We found that *MpmMn-SOD* also distributes in all the tested tissues, but the expression levels varied greatly among tissues, and the highest was in hindgut, followed by in fat body. Thus, *MpmMn-SOD* may mainly function in hindgut and fat body. The hindgut includes Malpighian tubule which plays an important role in detoxification and elimination of toxins (*Beyenbach, Skaer & Dow, 2010*). And fat body is one of the prime sites for antioxidant enzymes (*Kwang Sik et al., 2005*; *Yamamoto et al., 2005a*; *Yamamoto et al., 2005b*; *Yamamoto et al., 2005d*). Our result is similar to those on *Glossina morsitans* (*Munks et al., 2010*) and *Agrilus planipennis* (*Swapna Priya et al., 2011*), they both have significant SOD mRNA levels in fat body and hindgut. The great up-regulation of *MpmMn-SOD* in hindgut and fat body, in turn, indicated that these two tissues were important sites for resisting oxidative attack.

Mn-SOD has been considered a stress-responsive factor and its expression at the transcriptional and translational levels might be influenced by a variety of intracellular and environmental factors, including cold stress (*Fukuhara, Tezuka & Kageyama, 2002*). The *Mn-SOD* mRNA of the fall webworm *Hyphantria cunea* (*Kim et al., 2010*) and the bee *Apis cerana* (*Jia et al., 2014*) were highly increased at 4 °C. Similar result is also reported in oriental fruit fly *Bactrocera dorsalis* exposed to 0 °C (*Gao et al., 2013b*; *Gao et al., 2013a*). Our previous work found that *MpmMn-SOD* was one of the eight significantly up-regulated genes related to abiotic stress response in the transcriptomic data of the cold treated insects (*Tusong et al., 2017*). Here, we confirmed the expression profiling of *MpmMn-SOD* at 4 °C to detect the responsive pattern of the gene to cold stress. We found that *MpMn-SOD* mRNA level was very sensitively modulated by 4 °C cold stress. Within 0.5 h of 4 °C treatment, its expression increased to 9.9-fold of the control, and reached to 125-fold of the control at 1.5 h, strongly indicating that cold stress stimulates the expression of *MpmMn-SOD*. Our previous study on the *MpCu/Zn-SODs* showed that 4 °C stimulate the expression of *MpecCu/Zn-SOD* but not *MpicCu/Zn-SOD* (*Xikeranmu et al., 2019*). Compared with *MpecCu/Zn-SOD*, the *MpmMn-SOD* mRNA level was much higher than that of *MpecCu/Zn-SOD* which was 6.8-fold of the control 4 °C for 0.5 h, implying that *MpmMn-SOD* may play major role under cold stress. This was in consistent

with the location of MpmMn-*SOD* in mitochondria, where the electron-transport chain is responsible for a significant proportion of intracellular superoxide radical production (*Møller, 2010*). The rapid increase of the *MpmMn-SOD* levels under cold acclimation may reflect the adaptation of *M. punctipennis* to Guerbantonggut desert which is characterized with rapid and large temperature fluctuation. Similar result was found in the polychaete *Perinereis nuntia* treated with Cd (50 μg/L), in which *Mn-SOD* had a greater susceptibility than *Cu/Zn-SOD* (*Won et al., 2014*). It is noticeable that the cold expression profile of *MpmMn-SOD* under 4 °C presented as a stress-responsive type, which is characterized with drastic fluctuation during the cold treatment period, the first and second large peaks appeared at 1.5 h and 11 h of the cold treatment, which were 125-fold and 67.3-fold of the control respectively. The appearance of the second large peak looks like another round of cold defense is going on. These results may be interpreted as the cells timely adjusting the level of the enzyme to surrounding temperature to keep a relative intracellular balance, because stress-responsive expression is at the cost of inhibition of other genes expression. On the other hand, with the prolonging of cold stress, ROS increases again, and cells need to produce more MpmMn-SOD to deal with the excessive ROS. Our previous work (*Xikeranmu et al., 2019*) showed that there was a rapid increase of $O_2^{\bullet-}$ content in the beetle after an exposure at 4 °C for 10 h, which is consistent with this result in this work.

The anti-oxidative activity of MpmMn-SOD was examined by investigating the involvement of MpmMn-SOD in anti-oxidative stress by agar plate diffusion assay. The MpmMn-SOD-overexpressed bacteria had significant smaller diameters of death zones on agar plate than the control bacteria, demonstrating that MpmMn-SOD can significantly enhance cells tolerance to $H_2O_2$-mediated oxidative stress. *Jia et al. (2014)* observed similar results with ours, which shows the diameters of the death zones between the *Apis cerana* mMn-SOD-overexpressing bacteria and the control bacteria are obviously different under oxidative stressors. Our results showed that MpmMn-SOD indeed is an antioxidant enzyme that protect cells from oxidative damage.

Overexpression of MpmMn-SOD in BL21(pET32a-mMn-SOD) showed significant protective effect for the bacteria under cold stress, the survival curve of BL21(pET32a-mMn-SOD) at −4 °C was a convex type, while it was almost a negative line for the control bacteria, suggesting the cold resistance of BL21(pET32a-mMn-SOD) was significantly increased compared to BL21(pET32a). Further, we determined the SOD activity and the $O_2^{\bullet-}$ content of the transformed bacteria under −4 °C. Within our expectations, the enzyme activity during the cold treatment period was significantly higher than the control bacteria, and cold stress could stimulate SOD activity of the bacteria cells. Correspondingly, the $O_2^{\bullet-}$ content was significantly lower than the control during the cold period, indicating that the overexpression of MpmMn-SOD in *E. coli* cells enhanced cells ability to scavenge ROS thus to reduce oxidative damage under cold conditions. The changing trends of the relative increase of SOD activity and the relative decrease of $O_2^{\bullet-}$ content under cold stress was consistent, implying the more increase in SOD activity, the more decrease in $O_2^{\bullet-}$ content. The low levels of these two indexes at 4 h of cold treatment may due to a self-regulation of the cells to keep a relative subcellular balance in SOD gene expression. However, we noticed that at 0 h, the relative decrease of $O_2^{\bullet-}$ content was

roughly at the same level as at 2 h under cold conditions. This result may be explained as BL21(pET32a-Mn-SOD) had higher SOD activity than the control and it functioned well at room temperature.

Finally, we investigated the protective effect of MpmMn-SOD to BL21 (pET32a-mMn-SOD) under −4 °C. ROS accumulation can lead to membrane peroxidation and thus destroy cell structure and function (*Mittler et al., 2004*). Thus, we measured relative electrolyte leakage and MDA level in the bacteria cells. Within prediction, the plasma membrane leakage and MDA content in BL21 (pET32a-mMn-SOD) and BL21 (pET32a) both increased under cold stress, but the upward trend of the conductivity and MDA levels in BL21 (pET32a-mMn-SOD) were significantly lower than the control bacteria (Fig. 9). These results suggested that the damage degree to cell membrane under cold stress to the transgenic bacteria was significantly less than in control bacteria. Therefore, the high activity of MpmMn-SOD in the transformed bacteria should play its role in eliminating ROS, and thus preventing the membrane lipids from peroxidation. The present results agree with the work of *Kwon et al. (2010)*, who suggest that overexpression of SOD induced tolerance to membrane damage.

In this study the protective function of the insect mMn-SOD to cells under cold stress was effectively demonstrated in *E. coli* system. However, as prokaryotic cells lack mitochondria, the enzyme's activity may, to some extent, be affected, so it is better to use RNAi technology in future to test the function of the MpmMn-SOD in the desert beetle.

## CONCLUSIONS

In conclusion, the identified and characterized mitochondrial manganese superoxide dismutase gene (*MpmMn-SOD*) from the desert beetle *Microdera punctipennis* was tissue-specific, and cold inducible. It had anti-oxidative activity. The MpmMn-SOD overexpressed bacteria treated at −4 °C showed increased cold resistance. Its SOD activity and $O_2^{\bullet-}$ content at −4 °C were negatively correlated, implying that MpmMn-SOD act as a defense mechanism to mitigate cell damage caused by ROS under cold conditions. Accordingly, the MpmMn-SOD overexpressed bacteria weakened the plasma membrane damage caused by lipid peroxidation and kept better plasma membrane integrity under cold stress. Our findings provide basic data for further investigating the function, antioxidant mechanism and physiological responses of *Mn-SOD* gene in model species exposed to temperature changes. Obviously, additional studies based on gene knocking down technology are needed to gain further insights into the complex role of *Mn-SOD* gene in insect cold tolerance.

## ACKNOWLEDGEMENTS

We would like to acknowledge our fellow scholars, Fengjuan Zhang and Maimaitiaili Abudunasier for help collecting and identifying the insects used in this research.

## Funding

The research was funded by the National Natural Science Foundation of China (No. 31360527), a grant from Xinjiang Key Laboratory of Biological Resources and Genetic Engineering (No.XJDX0201-2014-03), and the Tianshan Cedar Project in 2017 2017xs20. The funders had no role in study design, data collection and analysis, decision to publish, or preparation of the manuscript.

## Grant Disclosures

The following grant information was disclosed by the authors:
National Natural Science Foundation of China: 31360527.
Xinjiang Key Laboratory of Biological Resources and Genetic Engineering: XJDX0201-2014-03.
Tianshan Cedar Project: 2017 2017xs20.

## Competing Interests

The authors declare there are no competing interests.

## Author Contributions

- Zilajiguli Xikeranmu conceived and designed the experiments, performed the experiments, analyzed the data, prepared figures and/or tables, authored or reviewed drafts of the paper, and approved the final draft.
- Ji Ma conceived and designed the experiments, analyzed the data, prepared figures and/or tables, and approved the final draft.
- Xiaoning Liu conceived and designed the experiments, analyzed the data, prepared figures and/or tables, authored or reviewed drafts of the paper, and approved the final draft.

## Field Study Permissions

The following information was supplied relating to field study approvals (i.e., approving body and any reference numbers):

The field site for the insect collection is free to access, so field permits were not needed.

## DNA Deposition

The following information was supplied regarding the deposition of DNA sequences:

Microdera punctipennis Mn-SOD gene sequences are available at GenBank: MK676072.

## Data Availability

The raw measurements are available in the Supplemental Files.

## Supplemental Information

Supplemental information for this article can be found online at http://dx.doi.org/10.7717/peerj.8507#supplemental-information.

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
