# Peer review of "Characterization of a Mn-SOD from the desert beetle Microdera punctipennis and its increased resistance to cold stress in E. coli cells"

_PeerJ, doi:10.7717/peerj.8507_

## Round 0.1 · original submission · Major Revisions

All concerns of the reviewers should be carefully addressed and the manuscript should be revised accordingly

·

Basic reporting

Manuscript for the most part written in clear and professional manner. However, there are some sentences that would need revision. It is suggested that the authors proofread the manuscript for some grammatical errors.

Experimental design

The identification and characterization of Mn-SOD from Desert Beetle and testing its cold-resistance property in E. coli is a novel study.
Major comments:
1) Methods section needs to be more detailed. Specifically, details on statistical analysis in methods only mention Tukey’s post-hoc test alone. It does not mention one-way ANOVA as its primary statistical test run by authors for identifying significant changes in their treatment data. Moreover, some of the experiments were analyzed by paired Student’s t-test which needs to be mentioned in the methods.
2) There is some detail on how relative electrical conductivity assay was performed and results analyzed in the supplementary file However, authors should explain the complete procedure in the methods section.
3) Though the overexpression data in E. coli is well put together and supports the hypothesis strongly, it would be great if authors could in near future show protection from cold stress by Mn-SOD via knock-down or knock-out models for Desert Beetle.

Validity of the findings

No comments

·

Basic reporting

This paper is a good follow-up of their previous publication based another two enzymes called “Cu/Zn-SODs”. Overall the paper submitted in PeerJ is well-written with sufficient intro about the background of the enzyme Mn-SOD, good results and informative discussion. The abstract is also clear and can summarize what they have found in this paper.
The cold-induced upregulation of MpmMn-SOD is interesting and I think the author has provided some results (figures and table, etc.) to support their claim, with appropriate references. However, I still have some concerns about the data shown in this paper and the consistency of the enzyme activity between the beetle and the E-coil. I will describe these concerns in the following parts.
In general, I think I recommend this paper to be published in PeerJ if the authors can address all my concerns.

Experimental design

The experiment design in this paper is similar to the Cu/Zn-SODs paper, except that the model enzyme used here is the MpmMn-SOD. Therefore, I think the research method described in this paper is systematic and solid-built. However, I still have some concerns about the experiment mentioned in this paper.
1. As the authors mentioned in the intro part (line 77), the average temperature of the soil the beetle would confront in Jan are -12 C to -5 C. However, the experimental temperature in this paper for beetle individuals was performed at 4 C. I am more concerned that the temperature difference will affect some of the data shown in the paper. I am also curious about the reasons that the authors wouldn’t perform these experiments in -5 C. Maybe the authors should add some sentences in the experiment part to address this. I did find out that the E-coil experiment was performed at -4 C. I think this temperature is much closer to the real temperature the enzyme will suffer.
2. The mRNA experiment shown in Fig. 4 is interesting but confused. In A it shows that the hindgut and fat body rank the first and second tissues that have more MpmMn-SOD expression. However, in B it only shows the expression of the overall mRNA in the beetle. In my opinion, B needs to show the different expressions in all the tissue listed in A, at least show the data in hindgut and fat body would be better. Also the data is compared to the head at 0h, which would give you the highest fold-change as the head has the lowest amount of MpmMn-SOD. But why you don’t compare it to the other 4 tissues listed in A?
3. Again, in figure 4B, at 11h the mRNA level seems to go back and it is very interesting. It looks like another round of cold-defense is going on? I recommend the authors could briefly explain this in results and discussion.
4. The SDS-PAGE experiment in Fig.5 A, there is a clear and strong up-regulation band in lane 2, located in ~20 KDa, what is this? It seems that the band disappeared in the lane 4. If lane 2 and lane 4 are both induced samples and the expression of pET32a-mMn-SOD wouldn’t affect the ~20KDa protein, should it be also on the lane 4, same position? Can you explain this?
5. Since the data in Fig.4 showed the experiment till 11 h (11 h shows a growing again and is interesting), it would be better that the same time data points (8h, 9h, 11h?) are shown and compared in Fig.6. Then you can compare the activity of the enzyme both in beetle and E-coil.
6. The Oxford-cup experiment in Fig. 7 is also interesting. Although the control showed a bigger circle compared to the MpmMn-SOD one at same H2O2 concentration, the circles in the control experiments are very dim and absent of a clear, sharp circle boundary. Why this happened? Can you explain?

Validity of the findings

Overall, I think the paper have shown some interesting findings. I already mentioned lots of comments in the previous section.

Additional comments

I think this paper is a good fit to PeerJ if the authors can address all my concerns lised above. Those comments are my main concerns about the paper.
Again, I have some minor revision, typo stuff for the authors, too.
1. The term BL21 in the abstact is very confused to me. I don't think that people would know it is a E-coil if they are not cell biologists. Maybe add it after "An E. coli system" in line 24, such as "An E. coli system (BL21) was applied ...."
2. Line 45, "reactive oxygen species" should be "ROS" as you mentioned the acronym before.
3. Line 53, "Superoxide dismutases" should be "Superoxide dismutases (SODs)", as you used SOD below.
4. Line 62, "MnSOD" should be "Mn-SOD".
5. Line 144, "MpmMnSOD" should be "MpmMn-SOD".
6. Sentence from Lines 301 to 302, please rewrite it.
7. Lines 403 and 411, two "futher"s seem weird. Could change the second futher to some other words, such as "last but not least"?
8. Line 542, the reference is wrong. The doi is same as the reference in line 538. Also I couldn't find the reference.
9. In SI material all the "datas" should be "data". "Data" is used as a plural noun in technical English, when the singular is datum.
10. Figure 5 labels A and B in figure are missing.

Overall, I recommend a major revision of the paper.

Reviewer 3 ·

Basic reporting

1. The paper requires corrections of a few errors in the text e.g. line 382 where the time required to reach 125-fold is missing.
2. The description and the y-axis title differ in figure 6B. Is it O2- content or O2- content reduction?
3. I would encourage the authors to include a section with commonly used acronyms.
4. Do insects have any other antioxidant enzymes? I believe it would enhance the knowledge of the reader if you were to mention in the introduction, others that might participate in the same process.
5. 2. Figure 2: Can you please add to your description, what the colors cyan, pink and black represent?
6. 3. What was the purpose of the analysis behind figure 3? Is there anything you can infer based on the classification about the behavior of the protein? Can any similarities be drawn with the enzyme version expressed by T. molitor? While you mention it in a few lines (231-235) in the results section, a bigger discussion about the different clades of enzymes in the introduction or results would help.
7. 5. Figure 6: Why does the SOD activity increase not correlate with a drop in O2- levels? What do you hypothesize as the reason for increase in O2- levels at 6h?
8. Does the over-expressed enzyme in BL21 bacteria localize to the mitochondria and if it does not, would that affect its effective activity?
Further, I encourage the authors to have the manuscript proof read.

Experimental design

No comment

Validity of the findings

In the experiments leading to Figure 6, it is mentioned that OD 595 readings were taken at the end of the assay. Would you mind sharing this as well? I would like to see how viability is affected by prolonged treatment to low temperatures and how this might affect the data.

Additional comments

No comment.

---

## Round 0.2 · Minor Revisions

Please address the remaining issues pointed out by the reviewers.

·

Basic reporting

A. The manuscript still needs proofreading. There are minor language errors throughout the manuscript that may create confusion. Some of the changes have been made to the "tracked changes" document and "rebuttal" which will be forwarded to the editor.

B. The authors have not added to discussion that testing the hypothesis with knocking down the enzyme in Desert Beetle is missing from current manuscript or is a weakness that they would address in future.

C. "To let the results be clearer to read Fig.6 was modified and supplied with the measured values of SOD activity (Fig.8A) and O2•- contents (Fig.8B), respectively (because OD595 readings were supplemented as in Fig6 in the revised version, the original Fig.6 was remarked as Fig.7, and so on for the following figures); The corresponding relative differences were given in Fig. 8C and Fig.8D, respectively".

Whatever the authors are trying to convey in this part of rebuttal is not clear at all. Kindly revise.

Experimental design

A. The authors have perhaps not understood the second reviewer’s comments about the mRNA experiment. In figure 4B, it is still unclear what the control at 0h is. In the correctly performed experiment, it should be mRNA level of the same tissue at 0h. Secondly, the labeling on top of bars (c, b, cd etc.) have not been explained in the legend. Please rectify these issues.

B. Please specify how the 4 concentrations for hydrogen peroxide for Oxford-cup experiment chosen.

Validity of the findings

No comment.

·

Basic reporting

The authors have addressed all my concerns very well. I think right now the revised manuscript meets all the criteria that PeerJ requires. I recommend the acceptance of the manuscript.

Experimental design

no comment

Validity of the findings

no comment

Additional comments

I only found some small typos in all the figure captions, which are located after the main pdf manuscript. Some sentences are repetitive. Please revise them in the final submitted version.

Reviewer 3 ·

Basic reporting

no comment

Experimental design

no comment

Validity of the findings

no comment

Additional comments

The authors have addressed all my comments satisfactorily.

---

## Round 0.3 · accepted · Accept

All remaining issues pointed by the reviewers were addressed and the revised version is acceptable now.